# Tribological Behavior of TiO_2_ PEEK Composite and Stainless Steel for Pediatric Crowns

**DOI:** 10.3390/ma16062420

**Published:** 2023-03-17

**Authors:** Ana Arieira, Sara Madeira, Flávio Rodrigues, Filipe Silva

**Affiliations:** 1Center for MicroElectroMechanical Systems—CMEMS, Campus de Azurém, University of Minho, 4800-058 Guimarães, Portugal; 2LABBELS—Associate Laboratory, 4800-058 Guimarães, Portugal

**Keywords:** coefficient of friction, wear rate, stainless steel, polyetheretherketone (PEEK)

## Abstract

Dental decay still presents a major health problem among children. Its treatment usually requires the use of stainless steel crowns. This study compares the wear behavior of 316 L stainless steel and polyetheretherketone (PEEK) composite under identical test conditions. The wear tests were conducted in a reciprocating ball-on-plate tribometer (Plint TE67/R) using alumina balls as a counterface and artificial saliva as a lubricant at 37 °C to faithfully mimic oral conditions. The coefficient of friction (COF) and specific wear rate (*k*) values were determined and SEM/EDS examinations were performed to identify the predominant wear mechanisms. Results showed that PEEK exhibited a significantly lower coefficient of friction (COF = 0.094 ± 0.004) and thus lower wear volume (Δ*V* = 0.0078 ± 0.0125 mm^3^) and higher wear resistance, with an average value of specific wear rate of *k* = 9.07 × 10^−6^ mm^3^N^−1^m^−1^ when compared to stainless steel (COF = 0.32 ± 0.03, Δ*V* = 0.0125 ± 0.0029 mm^3^, *k* = 1.45 × 10^−5^ mm^3^N^−1^m^−1^). PEEK was revealed to be a potential material for use in pediatric crowns due to its high wear resistance while overcoming the disadvantages associated with steel at both an aesthetic and biological level.

## 1. Introduction

Despite improvements in oral health in high-income countries, dental decay still presents one of the most widespread and significant public health problems among children [1]. According to the 2022 WHO Global Oral Health Status Report, over 514 million children have primary teeth caries [2]. A survey conducted by Public Health England in 2019 on the oral health of 5-year-old children showed that one in four children of that age have experienced dental caries [3].

Dental caries are caused by the interaction of bacteria from the plaque that forms on the surface of a tooth and sugary foods on tooth enamel [2,4]. Untreated dental caries can lead to odontogenic infections; these are a common dental emergency in children’s hospitals, highlighting the consequences of primary tooth decay [1,5]. These conditions can have an important impact on children’s health, nutrition, growth, and general health and quality of life [4,5]. Untreated primary tooth decay can lead to pain, tissue inflammation, infection, dental abscess, malocclusion, and chewing disorders [1,5,6].

In order to maintain the primary teeth in the dental arch preceding the eruption of the permanent teeth, a variety of restorative solutions and materials have been used in pediatric dentistry to provide full coverage restorations [7,8,9]. One of the most used methods and restorative materials in the treatment of tooth decay in children is stainless steel crowns (SSCs), renowned for their durability, long-term retention, and high clinical success rates in restoring larger carious lesions on primary molars [6,7,8]. SSCs are prefabricated, adapted to individual teeth, and cemented with a biocompatible luting agent. Despite the considerable amount of literature that supports the success of SSCs, they have some drawbacks. One major drawback is their poor aesthetic appearance [8]. Moreover, several studies revealed severe adverse effects of the ionic composition of SSCs in children (especially regarding the significant percentage of nickel), including the development of local inflammation or an allergic reaction to the systematic distribution of the ions as well as cytotoxic or genotoxic effects [10,11,12]. Several alternatives have emerged to address the use of metals and the unappealing appearance of dental restorations; these include pre-veneered stainless-steel crowns (PVSSCs), crowns composed of polymers (mainly PMMA), pre-veneered aluminum crowns, and prefabricated zirconia crowns, all of which aim to imitate the natural color of teeth. Nevertheless, the use of PVSSCs carries the potential risk of nickel allergy and sensitivity, while polymer and pre-veneered aluminum crowns have poor mechanical and wear resistance. Zirconia crowns are costly and necessitate extensive tooth preparation for placement [6,8,13,14].

More recently, and beyond just academic investigations polyetheretherketone (PEEK) has been used and already commercialized for several dental devices, including dental implants, abutments, healing caps, orthodontic braces, and denture prosthetic frameworks [15,16,17]. Further, PEEK composites [17,18,19], in particular TiO_2_ reinforced PEEK, have been proposed to overcome some wear limitations and achieve color improvement. A study conducted on the effect of nano-sized TiO_2_ addition on the tribological behavior of PEEK composite [19] using a pin-on-disc test revealed the superiority of rigid TiO_2_ nanoparticle-loaded PEEK in terms of specific wear rate compared to neat PEEK. Additionally, titanium dioxide structures are known for their superior mechanical properties and strong antibacterial action [18].

The purpose of this article is to carry out a comparative tribological study between PEEK composite reinforced with TiO_2_ nanoparticles and 316 L stainless steel to assess the potential of PEEK composite as an alternative material to stainless steel. This alternative is primarily for use in pediatric crowns due to its color and easier adaptation to the tooth due to its elasticity as well as its biological advantages, including non-cytotoxicity. According to studies in the literature about the mechanical superiority of TiO_2_ reinforced PEEK, it is expected that this material presents a good response to wear.

## 2. Materials and Methods

### 2.1. Materials

This study comprises the comparison of two materials of distinct nature: a conventional material used in pediatric dental restorations, 316 L stainless steel (SS), and PEEK composite reinforced with 20 wt%-TiO_2_ nanoparticles acquired from the Dental Direkt Company. The chemical composition (wt.%) of the stainless steel is Fe-62.7%, Cr-16.2%, Ni-9.6%, C-2.4%, Si-0.7%, Mo-2.1% and Mn-1.6%. The PEEK composite has the following elements in its chemical composition (wt.%): C-65.6% and Ti-14%. 

### 2.2. Sample Preparation

Samples of both materials (PEEK and SS) were cut into circular pieces (∅ 8 mm × 0.2 mm thickness) from commercial blocks. A thickness of 0.2 mm was used to simulate the thickness of the commercialized SS pediatric crowns to ensure that there is no bias caused by the difference in thickness regarding the results obtained from the different tests conducted on the different samples.

To confirm that the results were not impacted by the finishing process, all specimens underwent an identical process, similar to other studies [13,20]. Silicon carbide (SiC) papers with grits ranging from 800 to 4000 were used to polish the samples under running water. The final surface roughness of each material is presented in Table 1. The specimens were subjected to a 5-min ultrasonic bath in isopropyl alcohol.

A study conducted by Sampaio et al. on the influence of PEEK thickness on contact stress showed that the COF and the wear rate increased with decreasing PEEK thickness, resulting from the increased contact stress with the material [21]. Considering the importance of these factors on the long-term success of PEEK crowns, all samples, with 0.2 mm thickness, were cemented onto 2 mm thick zirconia samples in other to represent the conditions of the use of dental crowns, where the zirconia represents human teeth. The samples were cemented using the Bifix Hybrid Abutment, a universal luting composite, from VOCO. The zirconia discs were previously sandblasted with 150 µm alumina sand, at 2 bar, to create roughness on the surface, thus allowing improved cementation. During cementation, a weight of 2 kg was placed on top of each sample to ensure that there was the necessary load for the correct bonding of the material samples to the zirconia. Figure 1 represents the bonding of the SS and PEEK samples to the zirconia substrate.

Prior to the sliding wear testing, the hardness of each material was assessed using a nanoindenter (NanoTest—Micro Materials) equipped with a Berkovich diamond indenter type. Nine indentations at the maximum load of 50 mN for stainless steel and 100 mN for PEEK, with a dwell time of 10 s, were created in each specimen. The load-unloading cycles were carried out at a loading rate of 0.1 mN/s.

### 2.3. Colour Measurement

The shade and whitening values of the 20 wt%-TiO_2_ PEEK Composite and the conventional PEEK were measured for comparison purposes (Figure 1). The measurements were performed using a digital device, the Vita Easyshade, widely used in the area of dentistry. The final values were obtained using the Vita Classical shade guide and the Vita Bleached guide from the device.

### 2.4. Wear Testing

Test conditions were selected according to a previous study conduicted by Amanda et al. [20]. Reciprocating sliding tests were performed in a ball-on-plate configuration using a Plint TE67/R tribometer. The tests involved loading samples of SS and PEEK against 10 mm diameter alumina balls, as shown in Figure 2. Similar to other studies [22,23,24], alumina was selected as the counterpart material due to its increased and improved mechanical and chemical inertness qualities. The samples cemented to zirconia were carefully mounted on a 3D-printed PLA holder attached to the specimen support.

All wear tests were conducted in lubricated conditions at 37 ± 3 °C [25] using Fusayama–Meyer’s artificial saliva solution to reliably replicate human oral conditions, as performed in previous works [26]. The composition of the artificial saliva is indicated in Table 2. The pH was corrected to the range of normal human saliva pH values (6.2–7.6) [27].

To mimic clinical loads within the acceptable range, a normal load of 30 N was chosen [28] along with a reciprocating sliding frequency of 1 Hz and a 4 mm stroke length. Each test lasted 1 h and involved a total sliding distance of 28.8 m. To determine the average value, each test condition was performed three times. To quantify the running-in period and evaluate the friction coefficient in the steady-state friction regime, the COF was continually measured during sliding. 

A 3D optical profilometer (Filmetrics Inc., San Diego, CA, USA) was used to assess the profile of the wear tracks. The wear width and wear depth, measured using the profilometer, were used to calculate the wear volume of the samples. The wear track model used to calculate the wear volume is represented in Figure 3.

Empirical mathematical equations were used to perform the calculations, based on the assumption that wear tracks result from a perfect ball geometry. The calculation for the mid-zone area of the track was conducted according to the schematic representation illustrated in Figure 4. The total wear track volume was determined using the following equation:(1)$$\Delta V=L\times \left[\frac{1}{2}\times R^{2}\times 2\sin^{-1}\left(\frac{a}{R}\right)-\frac{b\times h^{\prime }}{2}\right]+\frac{\pi \times b^{4}}{64R}$$
where the Δ*V* is the total volume loss of the wear track in mm^3^, *R* is the radius of the alumina ball in mm, *a* is half of the width of the wear track (a=b/2) in mm, *L* is the stroke length in mm, and *h*′ is the height of the triangle in mm (see Figure 4B). This method has been reported in previous similar studies [29,30]. 

The following equation was used to calculate the specific wear rate (*k*):(2)$$k=\frac{∆V}{W\times L}$$
where Δ*V* is the worn volume in mm^3^, *L* is the total sliding distance in m, and *W* is the normal applied load in N.

After coating the worn samples with gold via sputter-coating, SEM/EDS characterization was conducted to identify the primary friction and wear mechanisms.

## 3. Results and Discussion

Regarding the shade and whiteness measurements of both PEEK samples, the results revealed that the 20 wt%—TiO_2_ PEEK composite presented a shade in the lighter tooth shade zone of the Vita Classical shade guide and an adequate whitening level; the conventional PEEK sample presented a significantly darker shade and a much higher value on the whitening scale. This is in line with the fact that it is a darker material. The Vita Bleached guide is composed of a scale from 1 to 29, where 1 represents the whitest shade and 29 the darkest. This guide is used to plan and monitor the tooth whitening processes. The results are presented in Table 3.

Figure 5 presents the typical evolution of the COF for both materials tested (SS and PEEK), obtained from the wear test.

The evolution of the COF of the thin disc samples (stainless steel and PEEK) was characterized by two distinct behaviors. In the case of the SS, the friction coefficient showed a lower value (about 0.075) up to 3 m, when it then increased to a value of 0.35 between the distance of 3 and 5 m. After this sliding distance, the stainless steel reached a steady-state regime with the COF value stabilizing around 0.32. The lower value of the COF until 3 m might be explained by the formation of an oxide film on the stainless steel. However, that oxide film was then torn, which allowed the alumina ball to completely touch the substrate, translating into higher values of COF (0.32 ± 0.03). This behavior of SS in wear tests has already been verified in previous studies [24].

Conversely, in the case of the PEEK composite, a well-defined stationary phase was achieved almost instantly at the beginning of the test; the COF value remained practically unchanged during both the running-in phase and in the steady-state regime. The friction coefficient of both materials was well stabilized at the end of the 1 h of sliding of the wear test.

Figure 6 illustrates the measured steady-state friction coefficient values for the reciprocating sliding of SS and PEEK against alumina while using synthetic saliva. As previously presented, the COF value for SS was around 0.32 ± 0.03, while the mean values for PEEK are much lower, at around 0.094 ± 0.004. The COF of SS is ≈3.4 times higher than PEEK. The values are in line with those in the literature [29].

The hardness values obtained from both materials are recorded in Table 4, with stainless steel having the highest value. The load and lubrication conditions for the investigated materials with varying hardness were the same across all experiments. A linear correlation between steady-state COF and hardness values could not be established, as the material with the highest hardness did not exhibit the highest COF. PEEK with the lowest hardness presented COF values ≈3.4 times lower than SS. This behavior has already been observed in other studies [21].

Figure 7 displays the 3D profiles of the wear track for both tested materials, as obtained through profilometry analysis. It should be noted that the figure only shows a segment of the wear track rather than the entire track, as, in the case of the stainless steel, the wear products obtained during the test and the crystallization of the saliva prevented the clear measurement of the track in its entirety. In the case of PEEK, even after gold plating, it was also not possible to read the complete track.

It should also be noted that these images may not reflect reality accurately since the depth of the wear track appears to be much greater due to the distinct numerical scaling in the different axes automatically implemented in the software. However, it is possible to observe, through the graphs of the track profiles of both materials (Figure 7B,D), that there is a considerable difference in the depth of the wear track between steel and PEEK; the depth of the stainless steel track is about 2.1 times greater than that of PEEK. While the mean depth of the wear track of the SS sample is 5.9 ± 0.3 µm, the mean depth of the wear track of the PEEK sample is considerably lower, at around 2.8 ± 1.1 µm. This difference is also reflected in the wear volume value of each material and is in accordance with specific wear rate values, as seen in Figure 8. The mean width of the wear track is similar for both materials; the SS track width (610 ± 79.4 µm) was only slightly greater than PEEK (583.3 ± 57.7 µm). It is worth mentioning that due to the X-axis compression that emphasizes possible small unevenness caused by the accumulation of wear products and saliva crystallization, the wear track may appear uneven. However, this does not represent the truth. 

Wear volume and specific wear rate are presented in Figure 8. PEEK showed a significantly lower wear volume (0.0078 ± 0.0125 mm^3^) and specific wear rate (*k* = 9.07 × 10^−6^ mm^3^N^−1^m^−1^), being ≈1.6 times lower when compared to SS. PEEK presents substantially higher wear resistance than SS. A similar study conducted by Jacobs et al. [31] revealed values in the same order of magnitude for the specific wear rate of PEEK compounds containing carbon fibers, glass fibers, PTFE, and graphite against alumina (*k* = 7 × 10^−6^ mm^3^N^−1^m^−1^). No specific wear rate values were found in the literature for validation purposes for TiO_2_ PEEK composite under the same test conditions as in the present study. Additionally, Gao et al. [32] reported a specific wear rate value of *k* = 21.32 × 10^−5^ mm^3^N^−1^m^−1^ for SS.

SEM micrographs depicting the morphological characteristics of the worn surfaces of the tested samples of stainless steel and PEEK are presented in Figure 9 and Figure 10, respectively. Additional SEM images of a portion of the SS and PEEK samples without catching the wear track were also obtained for comparison reasons (Figure 9A and Figure 10A). 

The worn surface of the SS denoted the formation of an outer layer of passive oxide with multiple detachments in a “puzzlelike” morphology, as well as adhered tribo-layers (Figure 9C area Z2). The development of these tribo-layers was ascribed to the following mechanism: in the presence of artificial saliva, the wear debris separated from the surfaces were compressed at the sliding interface, producing layers of clustered material that adhered to the plate and formed a substantial tribo-layer. The formation of the passive oxide layer has already been documented in previous studies [33]. Due to crack growth caused by the layers becoming thick and unstable, and during the sliding motion, these layers start to delaminate, revealing a damaged material surface that indicates the loss of material from the substrate underneath (Figure 9C area Z1). Similar to what occurred in comparable studies [34], the tribo-layer formed during the SS wear test did not add a protective layer to the material; this is in line with the high values of coefficient of friction and specific-wear rate obtained when compared to PEEK.

Regarding the chemical composition of the layers formed on the stainless steel specimen, EDS analysis confirmed the existence of components of artificial saliva (Na, Ca, P, and K) in significantly higher quantities in area Z2, where the tribo-layer is present, compared to area Z1 where this tribo-layer was removed, making the underneath material visible; this justifies the low values of saliva components and the high values of stainless steel components (Fe, Cr, C, Mo, Mn, Ni, and Si) in that zone. High contents of Fe and Cr, as well as lower Ni, Mn, and Mo content, also represent the typical chemical composition of the oxide layer of SS [33].

EDS analysis of the substrate outside the wear track confirmed the chemical composition of the stainless steel material without the presence of artificial saliva components; this revealed that the layers were only formed inside the wear track. Therefore, their development was assisted tribologically through the accumulation and adherence of saliva components with wear debris from the specimen. SEM micrographs of the PEEK surface revealed parallel grooves along the sliding direction resulting from abrasive wear against the hard ceramic counterface, interspersed with polished-looking smooth areas (as shown in Figure 10B); this represented an important role in the decrease of the mechanical component friction. These results are in accordance with the low steady-state friction coefficient values measured during the wear test when compared to the SS sample (Figure 6). EDS analysis from the surface of the subtract without the wear track revealed a clean surface, with only C and Ti present in its chemical composition (Figure 10A). In Figure 10C, the area Z1 of the wear track showed the same chemical composition as the clean substrate; however, areas Z2 and Z3 revealed a low percentage of Si and Al components. The presence of these components is not due to particles from wear on the alumina ball, but rather to particles from the used glue that could be contaminating the surface of the sample [35]. The entire sample was clean of artificial saliva components, which denotes that the wear mechanism was pure abrasion without the formation of tribological layers.

SEM micrographs of the worn surface of the alumina ball after sliding against the stainless steel (A) and PEEK composite (B) are presented in Figure 11. In the cases of the SS test, the alumina ball indicated the presence of adhered tribo-layers, which were confirmed by EDS analysis to be rich in saliva constituents (Na, Ca, P, and K). Components of the SS (Fe, Cr, Mn, Ni, and Mo) were also seen in the EDS analysis, which indicated that the wear materials of the SS sample were transferred onto the alumina ball. The alumina wear mark and the stainless steel counterface (Figure 11A) both have a rougher look which supports the obtained values of the steady-state friction coefficient shown by this tribo-pair being the highest (Figure 6).

Conversely, in the case of PEEK, the alumina ball exhibited a smooth surface with only slight abrasion, indicating a mild abrasion mechanism and little to no transfer of material from the composite counterface and artificial saliva, as shown in Figure 11B. This result is confirmed by the low specific wear rate obtained. EDS analysis also supported this observation, with components of the saliva (P and Ca) practically non-existent and disregarded. As there are no tribo-layers formed, the presence of these low-content salivary components can be justified by the crystallization of the artificial saliva after the test. Therefore, the PEEK composite presented a lower steady-state friction coefficient compared to the SS, and the specific wear rate was considerably reduced.

In summary, the current study used an in vitro simulation test setup, widely used and reported in the literature, to assess the wear behavior of a common restorative material, 316 L stainless steel, and PEEK composite, against alumina while mimicking oral conditions. 

The results of this study demonstrated that the PEEK material can be considered to be promising in the future of the dental field as a complementary or alternative choice to the traditional and established metal, stainless steel, due to its whitish color, high elasticity/adaptability, and high mechanical and chemical-resistant properties. Besides having higher wear resistance, PEEK offers a non-cytotoxic, electrically non-conductive, thermally insulating, and lightweight alternative to SS [15,16,17,36]. 

Nevertheless, this study presents some limitations. 

To mimic opposing teeth in a typical clinical scenario, it would be more appropriate to utilize natural enamel or dentin as the counterpart, instead of alumina. Furthermore, other factors, such as varying saliva compositions, pH levels [37], and temperature, along with varied cyclic loads and patient diets, can contribute to wear result variations [38]. Although these aspects do not appear to affect the findings of this study, it would be beneficial to perform additional tests accounting for these factors to complement this work.

## 4. Conclusions

In this study, the tribological behavior of thin sheets of the 316 L stainless steel and PEEK composite reinforced with nanoparticles of titanium were characterized and compared under the same test conditions.

Based on the results, the following conclusions can be deduced:The PEEK composite as a thin layer over a zirconia structure had lower COF and wear volume, and, consequently, better wear resistance when compared to stainless steel.The dominant wear mechanisms that occurred during the tribological test were abrasion in the case of PEEK and adhesion with the formation of oxide and tribo-layers in the stainless steel.According to EDS analysis, the presence of adherent tribo-layers resulted from a combination of saliva components and wear debris from the plate, with their formation being tribologically aided.In the specific case of stainless steel, the oxide and tribo-layers formed did not provide a protective layer against wear. This is in line with previous studies.

## Figures and Tables

**Figure 1 materials-16-02420-f001:**
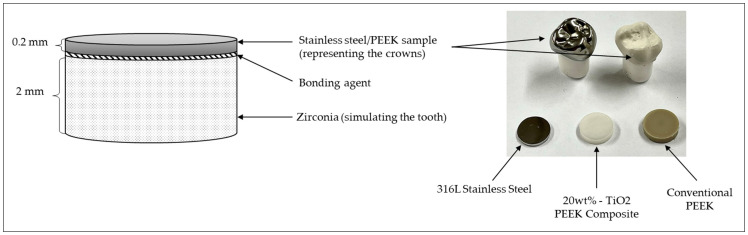
Representation of the stainless steel and PEEK samples glued to the zirconia discs and image of samples and tooth crowns of the respective materials (the image is not to scale).

**Figure 2 materials-16-02420-f002:**
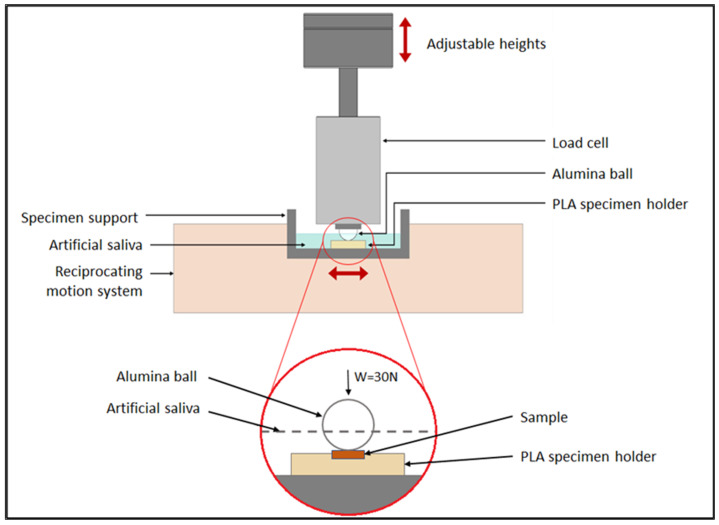
Schematic representation of the wear test.

**Figure 3 materials-16-02420-f003:**
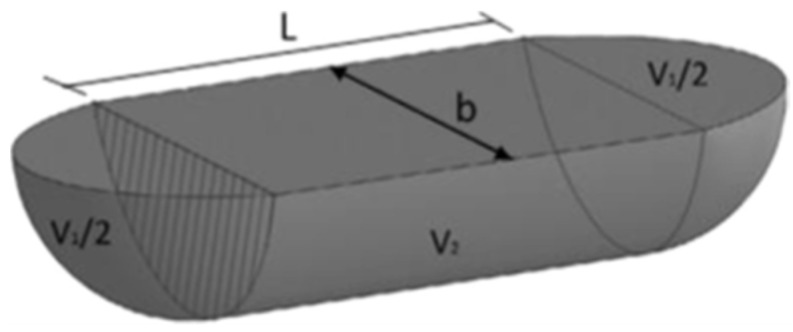
Wear track volume calculation model [29].

**Figure 4 materials-16-02420-f004:**
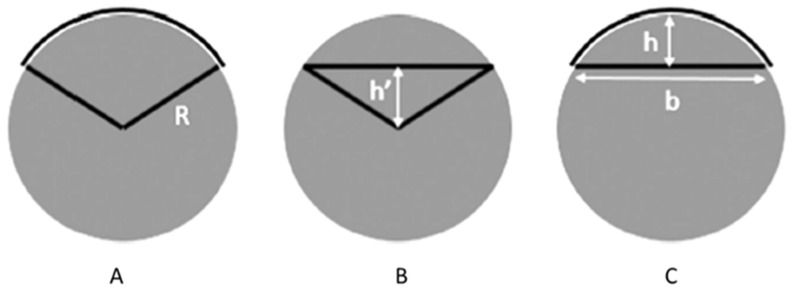
Calculation of the area of the mid-zone of the wear track: (**A**) area of the section, (**B**) area of the triangle, and (**C**) real area of the wear track [29].

**Figure 5 materials-16-02420-f005:**
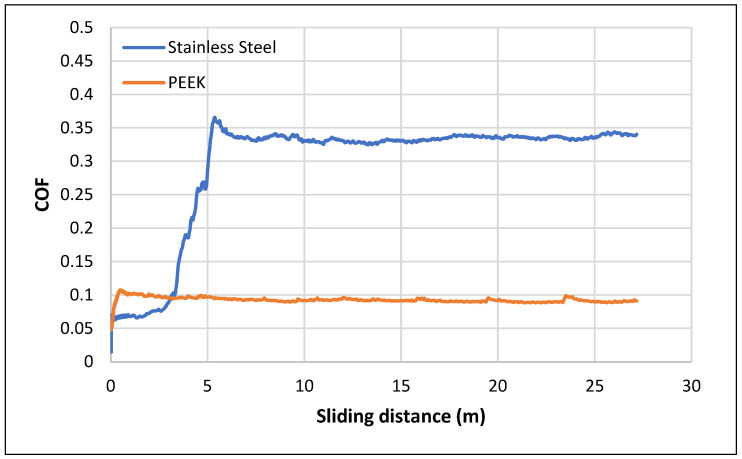
Evolution of the friction coefficient (COF) during sliding against the alumina ball in the presence of artificial saliva for the two tested (a sliding distance of 28.8 m was covered, equivalent to 1 h of sliding).

**Figure 6 materials-16-02420-f006:**
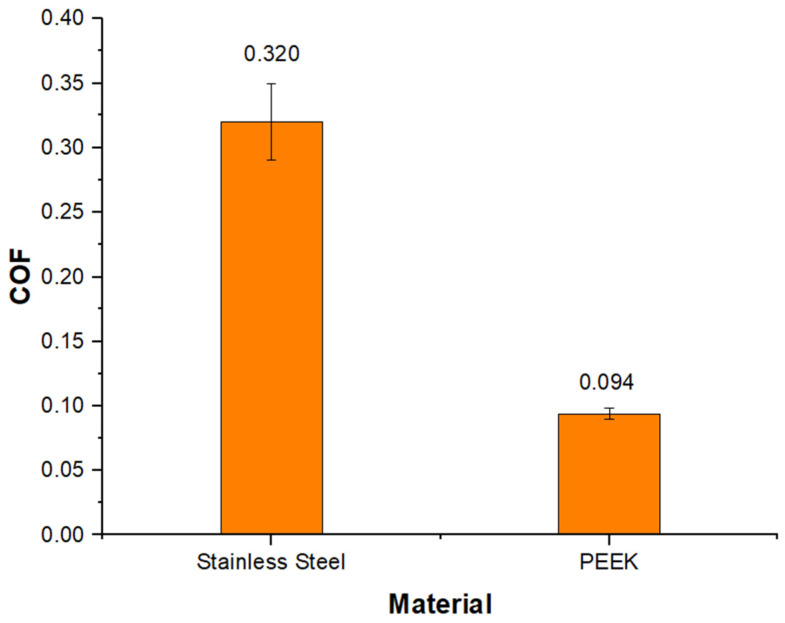
Steady-state coefficient of friction for stainless steel and PEEK against alumina in the presence of artificial saliva after 1 h of sliding.

**Figure 7 materials-16-02420-f007:**
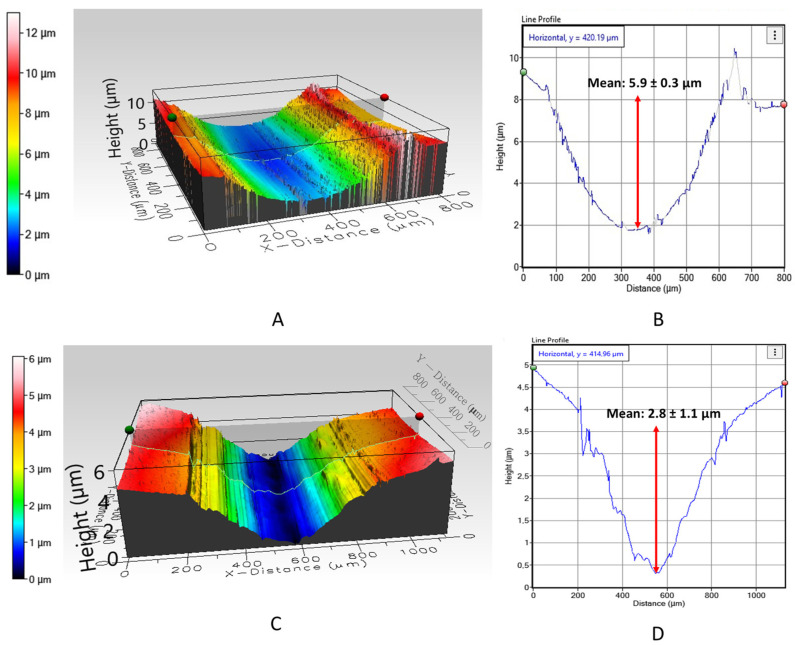
Optical profilometry scans of wear tracks of: (**A**), stainless steel, and (**C**), PEEK; and profiles of the wear track of: (**B**), stainless steel, and (**D**), PEEK.

**Figure 8 materials-16-02420-f008:**
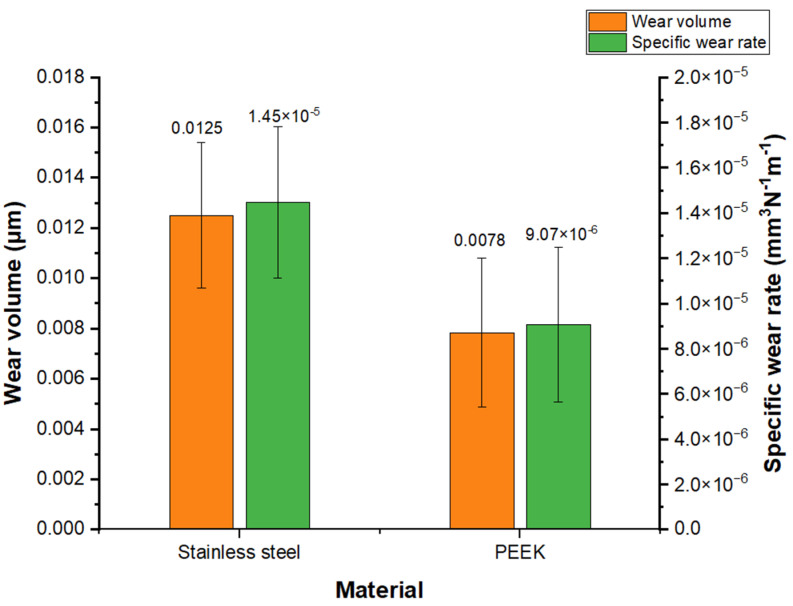
Wear volume and specific wear rate values for PEEK and stainless steel.

**Figure 9 materials-16-02420-f009:**
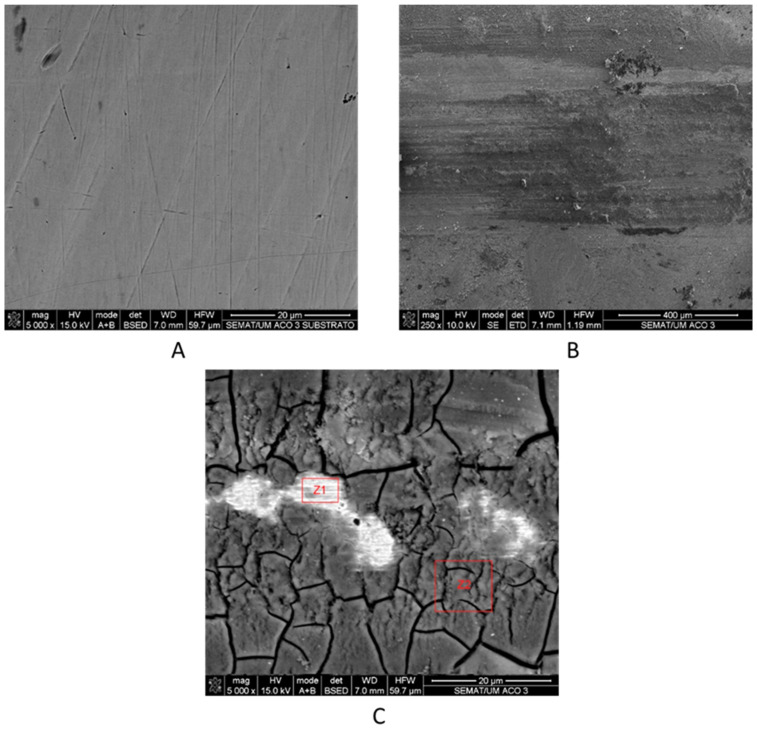
SEM micrographs of the: (**A**)—stainless steel sample surface without the wear track; and worn surface after 1 h sliding at a (**B**) 250× and (**C**) 5000× magnitude.

**Figure 10 materials-16-02420-f010:**
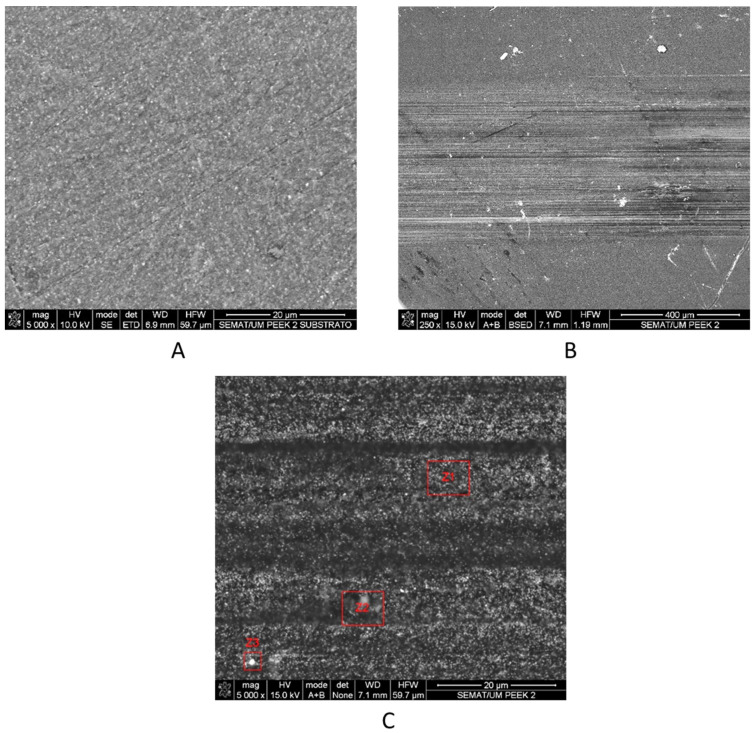
SEM micrographs of the: (**A**)—PEEK sample surface without the wear track; and worn surface after 1 h sliding at a (**B**) 250× and (**C**) 5000× magnitude.

**Figure 11 materials-16-02420-f011:**
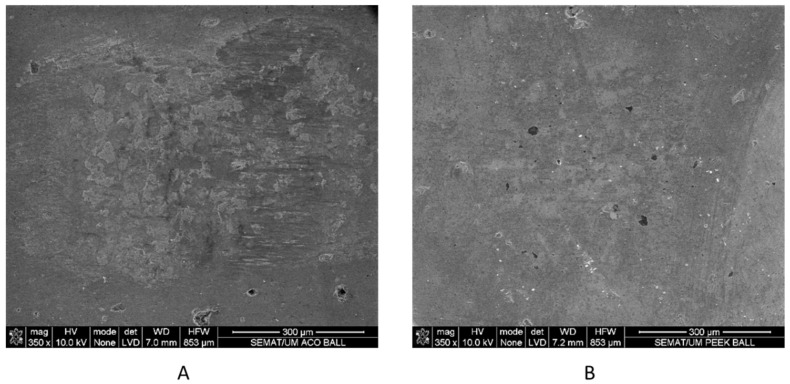
SEM micrographs of the alumina ball worn surface after 1 h sliding against (**A**)—stainless steel and (**B**)—PEEK.

**Table 1 materials-16-02420-t001:** Mean surface roughness of stainless steel and PEEK samples.

Material	Mean Surface Roughness (µm)
316 L Stainless Steel	0.677 ± 0.077
PEEK composite	0.229 ± 0.097

**Table 2 materials-16-02420-t002:** Fusayama–Meyer’s artificial saliva solution composition.

NaCl (g)	KCl (g)	CaCl_2_·2H_2_O (g)	NaH_2_PO_4_ (g)	Na_2_S·9H_2_O (g)	Urea (g)	Distilled Water (mL)
0.4	0.4	0.906	0.69	0.005	1	1000

**Table 3 materials-16-02420-t003:** Shade guide and bleach guide of TiO_2_ PEEK composite and conventional PEEK according to Vita.

	Material
20 wt%—TiO_2_ PEEK Composite 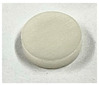	Conventional PEEK 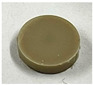
Vita Classical shade guide	A1	C4
Vita Bleached guide	4	20

**Table 4 materials-16-02420-t004:** Mean hardness of stainless steel and PEEK samples.

Material	Mean Hardness (GPa)
316 L Stainless steel	2.56 ± 0.26
PEEK composite	1.26 ± 0.32

## Data Availability

Data sharing not applicable.

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
