# Peer review of "Tribological Behavior of TiO2 PEEK Composite and Stainless Steel for Pediatric Crowns"

_materials, 2023, doi:10.3390/ma16062420_

Round 1
Reviewer 1 Report
The tribology of biomaterials is a very interesting and promising area of research. This area requires multidisciplinary knowledge to achieve better engineering applications. The authors have done good scientific work. But there are still some questions that need to be answered
(1)Is there currently a standard test method for biomaterial tribology? If so, how do the test methods in the text differ from those in the standard?
(2)Are there any special requirements for such biomaterials in the medical field?
(3)If there is a shock at the tribo-interface, will it affect your current test results?
Author Response
The authors are grateful for all the comments made to the manuscript, to which our greatest attention was devoted. All the reviewer’s suggestions were carefully analyzed in order to improve the quality of this work, by introducing additional information or carrying out local changes (highlighted in yellow in the manuscript). The changes marked as red by the “review” option of the word software concerns rewritten text phrases/expressions. Further information can be found within the attached word file.

Reviewer 2 Report
The work deals with a timely topic and the results can find practical application. However, the introduction should be expanded to demonstrate the validity of the research carried out since there are already studies of this material in the literature. The overall shape of the work is correct. The research problem posed has been solved. The tables are clear. Figures are correctly presented (except for Figure 7 in which comments are included in the appendix).
Detailed comments on the work are included in the appendix.

Author Response
The authors are grateful for all the comments made to the manuscript, to which our greatest attention was devoted. All the reviewer’s suggestions were carefully analyzed in order to improve the quality of this work, by introducing additional information or carrying out local changes (highlighted in yellow in the manuscript). The changes marked as red by the “review” option of the word software concerns rewritten text phrases/expressions. Further information can be in the appendix.

Reviewer 3 Report
Manuscript No.: Materials, 2253906
Date received February 23, 2023
Title: Tribological Behavior of TiO2 PEEK Composite and Stainless Steel for Pediatric Crowns
Authors: Ana Arieira, Sara Madeira, Flávio Rodrigues and Filipe Silva
According to the abstract, the paper investigated the PEEK to be a potential material for use in pediatric crowns due to its high wear resistance while overcoming the disadvantages associated with steel, both at an aesthetic and biological level. The wear tests were conducted in a reciprocating ball-on-plate tribometer using alumina balls as counter face. As a lubricant was used artificial saliva at 37ºC, faithfully mimic the oral conditions.
After carefully reviewing this paper, I recommend that it:
- The references are a lot but most of them are old. It must that 50% of the references be from the past 5 years so that your paper has citation value.
- From my point of view, this material “a conventional material used in pediatric dental restorations, 316L stainless steel (SS)“ is very old (have been widely used in pediatric dentistry for the last 50 years) and no longer of interest in pediatric dentistry in 2023.
- On line 112 you introduce table 4 immediately after table 1 “The obtained values are presented in Table 4. Moreover, this table is introduced only to the end of the paper.
- On line 121 you wrote “Test conditions were selected according to a previous work done by Amanda et al. 2020, entitled “Tribological Characterization of Dental Restorative Materials” [21].” In my opinion is enough to mention the number of reference and to introduce here the test conditions.
- On line 131 you wrote “All wear tests were conducted in lubricated conditions kept at 37 ± 3°C”, please let me know, how do you measure and maintain the temperature at 37 degrees?
- In the work is often said that all the results obtained by the authors were obtained in the references indicated in the bibliography. The immediate and natural question is what would be the novelty of this work and what interest it would have for the audience of Materials Journal.
- Please let us known about the motivations and goals of the paper efforts
- None of methods can be considered original.
- The work is quite messy and difficult to follow, please arrange it in a more scientific form.
- For all the above the presented work in this form needs improvement and my decision is that the paper to be accepted after major revision.
Author Response

(The authors gave the same response as above.)

Round 2
Reviewer 3 Report
the authors took into account the observations made previously